# Radiation Recall Pneumonitis Anticipates Bilateral Immune-Induced Pneumonitis in Non-Small Cell Lung Cancer

**DOI:** 10.3390/jcm12041266

**Published:** 2023-02-06

**Authors:** Sara Torresan, Gaia Zussino, Francesco Cortiula, Alessandro Follador, Gianpiero Fasola, Rossano Girometti, Lorenzo Cereser

**Affiliations:** 1Department of Oncology, Azienda Sanitaria Universitaria Friuli Centrale (ASUFC), 33100 Udine, Italy; 2Department of Medicine (DAME), University of Udine, 33100 Udine, Italy; 3Institute of Radiology, Department of Medicine (DAME), University of Udine, 33100 Udine, Italy; 4Department of Radiation Oncology (Maastro), GROW School for Oncology and Reproduction, Maastricht University Medical Centre, 6229 ET Maastricht, The Netherlands

**Keywords:** NSCLC, radiotherapy, immunotherapy, pneumonitis, radiation recall reactions, radiation recall pneumonitis, ICI-related pneumonitis, radiotherapy related pneumonitis, radiomics

## Abstract

Radiation recall pneumonitis (RRP) is a rare inflammatory reaction that occurs in previously irradiated fields, and it may be caused by various triggering agents. Immunotherapy has been reported to potentially be one of these triggers. However, precise mechanisms and specific treatments have not been explored yet due to a lack of data in this setting. Here, we report a case of a patient who received radiation therapy and immune checkpoint inhibitor therapy for non-small cell lung cancer. He developed first radiation recall pneumonitis and subsequently immune-checkpoint inhibitor-induced pneumonitis (IIP). After presenting the case, we discuss the currently available literature on RRP and the challenges of differential diagnosis between RRP, IIP, and other forms of pneumonitis. We believe that this case is of particular clinical value since it highlights the importance of including RRP in a differential diagnosis of lung consolidation during immunotherapy. Furthermore, it suggests that RRP might anticipate more extensive ICI-induced pneumonitis.

## 1. Introduction

Radiation recall pneumonitis (RRP) is an inflammatory reaction involving a previously irradiated lung area [1]. The pathological mechanism of RRPs is thought to be a drug hypersensitivity reaction in a tissue over-sensitized by previous radiation therapy (RT) [2]. The trigger of RRP might be any irritating agent that elicits an immune response, including immune checkpoint inhibitors (ICIs) [3]. ICIs are monoclonal antibodies inhibiting programmed cell death protein-1 (PD-1) and programmed cell death ligand-1 (PD-L1). PD-1 and PDL-1 are receptors located on the surface of cancer cells and T-cells, and they down-regulate the immune response [4]. PD-(L)1 signaling is also the mechanism used by tumor cells to evade the immune system [5]. By targeting PD-(L1), ICIs revert this mechanism unleashing patients’ immune systems against cancer cells [4].

ICIs represent the cornerstone in treating patients with non-small cell lung cancer (NSCLC) since they have been demonstrated to improve survival across almost all stages [6]. On the other hand, an unleashed immune system could cause immune-related adverse events (irAEs) such as immune-induced pneumonitis (IIP) [7,8].

Isolated cases and small retrospective series have reported the clinical and imaging features of RRP, considering it a completely different entity to IIP [9]. Data on the correlation between RRP and IIP are lacking.

Here, we report a case of a patient treated with post-surgical radiotherapy (RT), followed by ICI, who developed RRP and subsequent ICI-related pneumonitis.

## 2. Case Presentation

A 77-year-old man was diagnosed with stage IIIA NSCLC (adenocarcinoma) through mediastinoscopy on 31 July 2019. The patient was a heavy smoker (40 packs/year). He was diagnosed with major depressive disorder without any other significant comorbidity. After histological diagnosis and multidisciplinary discussion, the tumor was deemed resectable, and the patient underwent pulmonary lobectomy of the left lower lobe in August 2019. Pathological analyses of the surgical specimen confirmed the diagnosis and showed the following molecular profile: PDL-1 tumor proportion score (TPS) -determied at immunohistochemistry analyses/with SP263 antibody- was 90%; Kirsten rat sarcoma (KRAS) p.G13D mutation; no mutations of epidermal growth factor receptor (EGFR) (exons 18–19–20–21) and v-raf murine sarcoma viral oncogene homolog B1 (BRAF) (exon 15), and no rearrangements of anaplastic lymphoma kinase (ALK) and ROS proto-oncogene 1 (ROS1). The pathological stage was pT3pN1M0 (Stage IIIA per TNM 8th edition), and the resection was microscopically radical (R0).

Considering the patient’s age and clinical conditions, adjuvant chemotherapy was not administered. Instead, the patient received adjuvant RT, 54 Gy in 27 fractions (from October to December 2019). Figure 1 reports the adjuvant radiotherapy dose distribution and treatment sequence. The chest computed tomography (CT), performed at the end of the RT treatment, showed neither evidence of disease nor pulmonary toxicities.

In November 2020, the follow-up CT scan showed disease progression (new bilateral pulmonary lesions). In consideration of PDL-1 TPS expression (≥50%), the patient started treatment with the anti-PD1 antibody pembrolizumab. In February 2021, a routinely performed CT scan highlighted a new pulmonary consolidation, suspicious for disease progression (Figure 1), and pembrolizumab was halted. Other nodules were stable. A bronchial biopsy was performed to confirm disease progression, but it revealed post-radiotherapy scarring without any evidence of tumor cells.

Considering that RT was administered more than one year before, the bronchial biopsy results, and the fact that the consolidation appeared in the prior RT field, RRP was diagnosed. Considering that immunotherapy was the triggering agent of RRP, despite not being a classical IIP, the disease control was achieved; at the preference of the patient, pembrolizumab was halted and follow-up started. Oral corticosteroids were started to treat RRP, and they were tapered over the following eight weeks. The subsequent follow-up CT scan (May 2021) showed a partial resolution of the abnormalities in the left lung while revealing new, bilateral, and patchy airspace consolidations, consistent with grade (G) 2 (per CTCAE v. 5.0) organizing pneumonia (OP), possibly induced by ICI (Figure 1). Disease control was confirmed. The patient was symptomatic only for G1 dyspnea. Pulmonary function tests were performed, showing a moderate reduction of transfer capacity of the lung for carbon monoxide (TLCO). Blood tests showed no signs of infection (complete blood count, erythrocyte sedimentation rate, and C-reactive protein were only slightly altered). In consideration of the timing from the last dose of ICI, a negative initial workup for infections and a mild-symptom bronchial biopsy were not performed. Since the patient was already on steroids from the previous diagnosis of RRP, no further treatment was started. Symptoms progressively resolved in the subsequent days and the CT scan performed at 6 weeks (July 2021) showed a partial resolution of the pneumonia, with no new pulmonary findings. In November 2022, the patient was still alive and on follow-up, with no evidence of progressive disease, leading to a PFS of 24 months. The pneumonitis was completely resolved.

## 3. Discussion

We present the case of a patient developing RRP, probably triggered by ICI and subsequent bilateral IIP, despite the steroids being administered for RRP and ICI being halted. In our case, pulmonary toxicity correlated with disease control. Pulmonary toxicity, RRP, IIP, and infectious pneumonitis represent major issues in patients with NSCLC treated with immunotherapy and/or RT. Severe (grade ≥ 3 per CTCAE v. 5.0) pulmonary toxicities occur in about 20% of patients receiving RT for stage III NSCLC [10]. The incidence of grade ≥ 3 pneumonitis in patients receiving chemoradiation for stage III NSCLC ranges from 3.6% in randomized controlled trials (RCTs) to 12.4% in real word scenarios [11,12]. The typical time of occurrence of pneumonitis from RT initiation is 3–12 weeks, but some reviews report RPs up until six months after RT completion [13]. Established risk factors for RP include dose per volume, location of the tumor, older age, respiratory comorbidities, and concurrent chemotherapy [14,15,16].

In the PACIFIC trial, the cumulative incidence of IIP and radiation pneumonitis was 33.9% (3.4% grade ≥ 3) in the durvalumab arm and 24.8% (2.6%) in the placebo arm [17]. Real-world data showed an incidence of grade ≥ 3 pneumonitis of around 7–14% [18].

In RCTs involving patients with stage IV NSCLC, pneumonitis was one of the most severe immune-related adverse events (irAEs), occurring in 1.3% to 5.9% of patients [19,20,21]. Risk factors associated with ICI-induced pneumonitis are previous RT thoracic treatment and prior lung injury or disease [3]. Typically, immune-induced pneumonitis is experienced 2–4 months after the initiation of ICI [22]. However, some cases have been reported even >6 months after the initiation of immunotherapy, complicating the differential diagnosis between RRP and IIP [23]. Our patient developed ICI-induced pneumonitis 6 months after initiating immunotherapy and 2 months after withdrawal from it. This is not the most frequent timing, but it is a common experience in clinical practice, and it has been reported in the literature [24,25]. RRP may be triggered by chemotherapy, target therapies, and ICI [2,26]. The real incidence of RRP is unknown, and overall, radiation recall reactions—not limited to the lung tissue—are reported in 7% of patients receiving systemic treatment after RT and up to 18.8% of patients with NSCLC receiving ICIs [27]. ICIs enhance the local immune response in lung parenchyma, promoting inflammation in previously injured areas, thus triggering RRP [28]. RT promotes inflammation in the lung parenchyma through several mechanisms (Figure 2) [29]. A case series suggested that RRP might be associated with IIP, as in the case we are presenting [27]. This raises the question of whether RRP and IIP are two different pathological entities or different presentations of the same disease.

Whereas standard recommendations for RRP treatment are lacking, precise guidelines for treating IIP exist. Steroids usually lead to the resolution of the symptoms and radiological findings. In addition, withdrawal of the suspected triggering drug is recommended [23]. In some case reports of ICI-triggered RRP, ICI treatment was resumed following the resolution of the RRP, and no toxicity recurrences were reported [29].

Data on the correlation between RRP and good outcomes are controversial [27,30]. In the case we have presented, the RRP and subsequent IIP were associated with disease control, suggesting a competent anti-tumor immune activation. The development of RRP does not seem to impact prognosis; however, only sparse data about survival and RRP have been reported so far, and a different cause of RRP might also imply a different prognosis [31].

The differential diagnosis of IIP and RRP from other causes of pneumonitis or disease progression is of the utmost importance to administer the correct treatment and avoid unnecessary ICI interruption. Differential diagnosis may be challenging, particularly if the patient has received both RT and ICI and the lung infiltrates are localized and unilateral. Table 1 summarizes chest CT findings, clinical findings, and different workups for differential diagnosis between tumor progression, infection, ICI-related pneumonitis, RRP, and radiation-induced pneumonitis. Cultures on bronchoalveolar and biopsy are the gold standard for infection or PD diagnosis, respectively. However, these are burdensome procedures for patients, and the results are not immediately available. Thus, radiological features (along with clinical and laboratory findings) are key to a correct diagnosis. Organizing pneumonia (OP) is the most common CT pattern of RRP, consisting of confluent consolidation or ground-glass opacities with air bronchograms and bronchial dilatation and distortion, with diffuse or patchy distribution, affecting the prior RT fields [32]. Less common CT patterns include non-specific interstitial pneumonia (NSIP), hypersensitivity pneumonitis (HP), and acute interstitial pneumonia (AIP) [26]. Pulmonary infection mainly appears as consolidation and/or ground-glass opacity with air bronchograms, usually with an extension/distribution that differs from the irradiated field [33]. ICI- and RT-related pneumonitis share many CT features, the OP pattern being the most frequent presentation for both entities, followed by NSIP and HP patterns [34]. Unlike RT-related pneumonitis, including RRP, ICI-related pneumonitis tends to be bilateral, more likely to involve more lung lobes, and less likely to have well-demarcated borders [35].

Radiomics, a quantitative imaging analysis tool, has shown promising results in identifying the predominant pneumonitis etiology in patients receiving both RT and ICI [35,36]. Future studies are needed to confirm these preliminary results and make radiomics available at the bedside.

**Table 1 jcm-12-01266-t001:** Main chest CT findings, clinical findings, and suggested additional exams for differentiating radiation recall pneumonitis from tumor progression/recurrence, infection, immune checkpoint inhibitor pneumonitis, and radiation-induced pneumonitis [23,26,33,34,37,38,39,40].

	Chest CT Findings	Clinical Findings and Onset Time	Other Exams/Tests
Tumor progression/recurrence	Sequential enlargement of the mass/consolidation at the primary site; Enlargement occurring >12 months after RT completion; Disappearance of linear margins; development of bulging margins; Loss of air bronchogram; Increased soft tissue components.	Most local recurrences of malignancy occur within the first 3 years.	18F-FDG-PET/CT demonstrating increased uptake Close follow-up with CT TBB (especially in high-risk patients)
Infection	Consolidation and/or ground-glass opacity with air bronchogram, Usually occurring outside the prior radiation field and/or not respecting the boundaries.	In the non-neutropenic patient, infection presentation, whether viral or bacterial, is similar to that in patients with no cancer.	Laboratory findings, e.g., blood culture, sputum culture, urinary antigen test, and sensitivity tests. Response to appropriate therapy
ICI-related pneumonitis	Several distinct radiologic patterns: OP is the most commonly seen, followed by NSIP and HP patterns. Usually bilateral, with involvement of >1 lobe, particularly lower lobes. Less likely to have well-demarcated borders.	Extremely variable onset time (fimmu). More frequent at around 3 months after ICI starts. Variable clinical presentation, ranging from asymptomatic to life-threatening respiratory compromise, possibly leading to death.	BAL: increased lymphocyte count, with CD4+ T-cells predominance TBB: inflammation and lymphocyte infiltration
Radiation recall pneumonitis	Consolidative or ground-glass opacities closely corresponding to the prior radiation field, with a well-defined linear demarcation from the adjacent normal lung.	Patient presentation ranged from asymptomatic to severely symptomatic, with non-productive cough and dyspnea. Median time interval between RT completion and IT-induced RRP onset: 450 days (range, 231–1859 days). Median time interval between IT initiation and RRP onset: 61 days (range, 4–520 days).	No well-established consensus on RRP diagnosis and treatment. Worth considering: treatment timeline, clinical symptoms, physical examination, radiological images revision, and TBB
Radiation-induced pneumonitis	Usually occurring in the radiated lung. Acute phase radiation pneumonitis: focal, diffuse, or non-uniform GGO and/or consolidative opacities in the high-dose region of the RT field, usually with sharp demarcation. Possible evolution into OP and CEP pattern: Consolidations can coalesce, with relatively sharp borders conforming to the radiation field rather than to anatomic boundaries. Chronic phase radiation fibrosis, in case of severe injury: Conventional pattern: irregular consolidation + volume loss + architectural distortion + traction bronchiectasis in the radiation field. Mass-like pattern: focal consolidation confined to the site of the original tumor.	The acute phase starts 4–12 weeks after RT completion The chronic phase commonly shows slow progression in the 6–12 months after RT completion, with stabilization within 2 years.	18F-FDG-PET/CT demonstrating increased uptake Close follow-up with CT TBB (especially in high-risk patients)

CEP = chronic eosinophilic pneumonia; CHT = chemotherapy; CT = computed tomography; 18F-FDG-PET/CT = 18F-fluorodeoxyglucose positron emission tomography/computed tomography; GGO = ground-glass opacity; HP = hypersensitivity pneumonitis; ICI = immune checkpoint inhibitors; IT = immunotherapy; NSIP = non-specific interstitial pneumonia; OP = organizing pneumonia; RRP = radiation recall pneumonitis; RT = radiation therapy; TBB = transbronchial biopsy.

## 4. Conclusions

The recognition of RRP can be challenging, especially when it occurs years after the completion of radiotherapy, since it may mimic other lung conditions, such as tumor recurrence and pulmonary infection. The existence of clear pathological differentiation between RRP- and ICI-induced pneumonitis remains a matter of debate. This case is of particular clinical value since it highlights the importance of including RRP in the differential diagnosis of lung consolidation during immunotherapy. Furthermore, we showed how RRP might anticipate more extensive ICI-induced pneumonitis, advocating different surveillance and management programs for these patients.

## Figures and Tables

**Figure 1 jcm-12-01266-f001:**
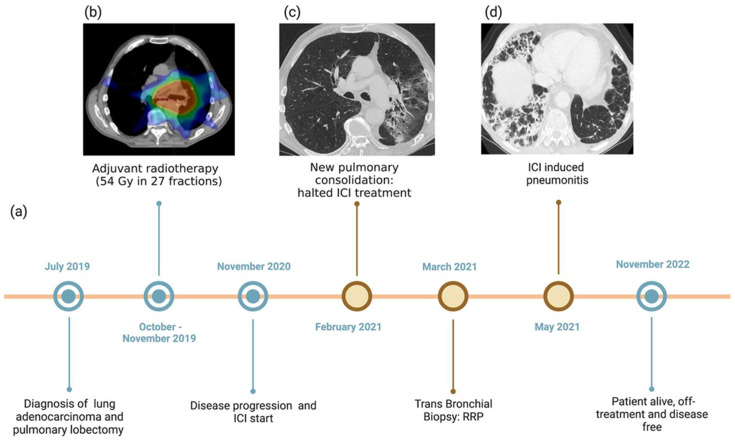
Overview of treatments received and diagnoses of RRP and ICI-induced pneumonitis, with corresponding CT findings. (**a**) Timeline showing the patient’s clinical history. (**b**) CT scan showing the radiation field in the left perihilar lung zone. (**c**) Radiation recall pneumonitis: CT image with lung windowing on the axial plane shows a broad, inhomogeneous left lung region of consolidation, ground-glass opacities, bronchial dilatation, and parenchymal distortion. Such findings are confined to the previously irradiated field, with a well-defined demarcation from the adjacent normal lung. No nodules, soft tissue masses, or ancillary signs of infection were detectable. (**d**) ICI-induced pneumonitis: CT image with lung windowing on the axial plane shows partial resolution of the perihilar left lung signs of radiation recall pneumonitis while revealing new, multiple, bilateral, and patchy airspace consolidations with predominantly perilobular and subpleural distribution, along with widespread air bronchogram signs and dilated bronchi. These findings define a bilateral, extensive organizing pneumonia pattern.

**Figure 2 jcm-12-01266-f002:**
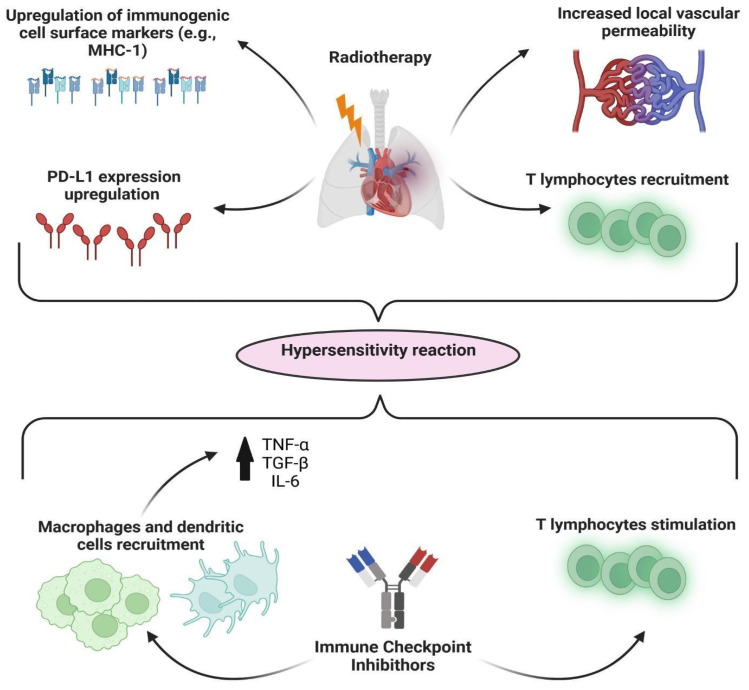
Possible mechanisms for inducing RRP. RT promotes a latent pro-inflammatory state through: (i) expression up-regulation of immunogenic cell surface markers (e.g., MHC-1), making the tumor cells more susceptible to CD8+ T-cell recognition and destruction; (ii) increased T-cell recruitment, lowering the inflammatory response threshold; (iii) PD-L1 expression up-regulation, leading to irradiated lung tissue hypersensitivity and anti-PD-1 blockade overreaction; (iv) increased local vascular permeability and proliferative changes. ICIs stimulate exhausted tumor-infiltrating lymphocytes and recruit new ones, leading to the recruitment of innate immunity cells (macrophages and dendritic cells) and increased production of inflammatory cytokines (TNF, IL-6, TGFβ, IL-4), evoking a hypersensitivity reaction in such a pre-sensitized tissue. MHC-1 = major histocompatibility complex type 1; PDL-1 = programmed cell death ligand-1; TNF-α/β = tumor necrosis factor; IL-6 = interleukin-6.

## Data Availability

Not applicable.

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
