# Peer review of "Radiation Recall Pneumonitis Anticipates Bilateral Immune-Induced Pneumonitis in Non-Small Cell Lung Cancer"

_jcm, 2023, doi:10.3390/jcm12041266_

Round 1

Reviewer 1 Report

This case report revealed a patient who received radiation therapy and immune checkpoint-inhibitors therapy for Non Small Cell Lung Cancer, who subsequently developed radiation recall pneumonitis and immune-checkpoint inhibitors related pneumonitis. This clinical case is interesting and has the potential appealing to Radiation Oncology community. The following issues should be resolved.

1.Both the abstract and introduction is rough.  Thoughts do not always follow each other logically. In abstract, authors only reported what they do in this study and no any introduction of ICI-related adverse reaction and RRP. Similarly, the Introduction section needs to be revised fully to elaborate the contents of this study.

2.The presentation of the case is poor.  Further details are required, for example, why patient choosed Pembrolizumab?

3.Figure 2 is missing.

4.In Discussion section, authors should summerize the literatures of radiation recall pneumonitis and discuss what is the advancement of this study.

5.Please elaborate the treatment and the specific prognosis of radiation recall pneumonitis.

6.English should be chekced during revisions.

Reviewer 2 Report

The case report describes a quite rare event that is radiation recall pneumonitis (RRP) and immune-related pneumonitis (IR-pneumonitis) in the same patient with non-small cell lung cancer (NSCLC). Overall, the text is well written and presented in a logical and structured way. However, I have some remarks that will hopefully improve the manuscript.

1.       Language: In general, English is understandable and correct. I have no major comments in that field.

2.       Major comments:

-          The diagnosis of IR-pneumonitis is well explained, but I have some concerns about how the diagnosis of IR-pneumonitis was established. The authors mentioned that it was revealed on CT scan and that the patient was asymptomatic, but what about other tests? According to the guidelines (for example, ESMO guideline Haanen et al. 2022) there are some baseline investigations which should be performed. In my opinion, the authors should provide the blood/sputum etc. test results or explain in detail why those tests were not performed. Otherwise, the diagnosis of IR-pneumonitis is doubtful.

-          In the discussion section, IR pneumonitis has been said to occur at the beginning of treatment, but sometimes even >6 months after the initiation of immunotherapy. In the described case, it appeared 5 months after withdrawal from immunotherapy. Therefore, this diagnosis is questionable, and the authors should provide some data on such a late occurrence of IR-pneumonitis and better explain how they diagnosed this patient.

-          Could the authors explain why pembrolizumab was not resumed when they excluded disease progression and toxicity?

-          The main educational value of this manuscript is the differential diagnosis in patients with NSCLC between entities: RRP, IR-pneumonitis, infectious pneumonitis, and radiation-induced pneumonitis. The authors provide some information regarding this subject in the discussion, but it is not very clear. I suggest putting this information in the table with some hints about mandatory tests in differential diagnosis between these entities.

3.       Minor comments:

-          All of the abbreviations (such as PDL-1, EGFR, etc.) should be explained .

-          The grade (according to the grade (per the Common Terminology Criteria for Adverse Events) of reported events should be specified.

-          Explanation of abbreviations in Figure 2 is missing.

Round 2

Reviewer 2 Report

Thank you for addressing all the remarks. I found this paper very helpful in clinical practice. In my opinion, the article is ready for publication.